# Dropping the Ball? The Welfare of Ball Pythons Traded in the EU and North America

**DOI:** 10.3390/ani10030413

**Published:** 2020-03-02

**Authors:** Neil D’Cruze, Suzi Paterson, Jennah Green, David Megson, Clifford Warwick, Emma Coulthard, John Norrey, Mark Auliya, Gemma Carder

**Affiliations:** 1World Animal Protection, 222 Gray’s Inn Road, London WC1X 8HB, UK; suzipaterson@hotmail.co.uk (S.P.); Jennahgreen@worldanimalprotection.org (J.G.); 2Wildlife Conservation Research Unit, Department of Zoology, University of Oxford, Recanati-Kaplan Centre, Tubney House, Abingdon Road, Tubney, Abingdon OX13 5QL, UK; 3Department of Natural Sciences, Manchester Metropolitan University, All Saints Building, All Saints, Manchester M15 6BH, UK; D.Megson@mmu.ac.uk (D.M.); E.Coulthard@mmu.ac.uk (E.C.); J.Norrey@mmu.ac.uk (J.N.); 4Emergent Disease Foundation, Suite 114 80 Churchill Square Business Centre, Kings Hill, Kent ME19 4YU, UK; cliffordwarwick@gmail.com; 5Zoological Research Museum Alexander Koenig, Department Herpetology, Adenauerallee 160, 53113 Bonn, Germany; mark.auliya@ufz.de; 6Department of Conservation Biology, Helmholtz Centre for Environmental Research GmbH—UFZ, 04318 Leipzig, Germany; 7Brooke, 2nd Floor, The Hallmark Building, 52–56 Leadenhall Street, London EC3M 5JE, UK; gemmacarder@yahoo.co.uk

**Keywords:** exotic pet, *python regius*, reptile, social media, wildlife trade

## Abstract

**Simple Summary:**

Ball pythons (family Pythonidae) are a relatively small species of snake found in west and central Africa. They are popular across the world as exotic pets, particularly in Europe and North America. Snakes are wild animals (i.e., non-domesticated) and have specific requirements for captive living. If they are housed in unsuitable conditions, it could negatively affect their health and wellbeing. Our study aimed to review the housing provided for this species by breeders and sellers advertising their snakes at exotic pet expositions and on YouTube. We assessed how much water, shelter and floor material were provided, as well as hygiene levels, and how much room the snakes had to move. We based our assessment on guidelines provided by the Royal Society for the Protection of Animals (RSPCA), the world’s first Animal Welfare charity). We found that most of the housing conditions we observed did not meet minimum recommendations. We also found that breeders and sellers did not provide adequate information for new pet owners detailing how to look after their snakes appropriately. We recommend that more research is required to help inform and improve guidelines for keeping snakes in better captive conditions, and that breeders and sellers should provide more guidance for pet owners, to stop Ball pythons kept as exotic pets from suffering.

**Abstract:**

Ball pythons (family Pythonidae) remain a commonly exploited species, readily available for purchase in North America and Europe. We assessed the housing conditions of more than 5000 Ball pythons across six exotic pet expositions and 113 YouTube videos. We scored provisions for hygiene, mobility, shelter, substrate and water provision, based on the Royal Society for the Protection of Animals (RSPCA) minimum guidelines. We found most entities involved in this commercial enterprise are not providing housing conditions that meet the minimum welfare recommendations for Ball pythons, either publicly or privately. We found that breeders and vendors typically utilised small and highly restrictive enclosures, with dimensions that prevented occupants from extending their bodies to full and unrestricted natural length. Our study also highlights that most vendors are not providing adequate written husbandry guidance to potential consumers, either at exotic pet expositions, on their commercial website, or on associated social media pages. Furthermore, our study also indicates that most potential consumers may themselves be unable to recognise unsuitable housing conditions that do not meet minimum animal welfare standards for Ball pythons. We suggest that more consistent guidance, adherence to agree principles and more potent operating models that are formally incorporated into relevant legislation would greatly aid existing and future efforts to safeguard animal welfare in this regard.

## 1. Introduction

For millennia, demand for exotic pets has been part of human culture [1]. A diverse range of wild animals was used for human entertainment and companionship in Ancient Egyptian [2], Greek, and Roman culture [3]. Today, trade is booming [4], and influenced by modern factors such as demand, infrastructure, and accessibility [5]. For example, widespread demand for reptiles as exotic pets is a relatively recent phenomenon, with this taxonomic group only becoming popular since the 1940s [6] and intensified (on a commercial scale) in subsequent decades [7]. Yet, this rise in popularity has now grown to the extent that they are currently thought to represent the second most species-rich vertebrate class, after birds [5] and fishes [8,9,10] in the international exotic pet trade.

Reptiles, like other taxonomic groups utilised as exotic pets, may be sourced directly from the wild, taken from the wild as juveniles (or in some cases eggs), or bred/born in captivity [5]. Irrespective of how they are sourced, the exotic pet trade can impact negatively on the welfare of the reptiles involved at all stages from “source” to “sink” (i.e., during collection, transport, and private ownership) [11,12]. One study involving self-declared mortalities among reptile breeders and keepers attending hobbyist events suggested a first-year rate of 3.6% [13], whereas another study involving supply versus resident populations of reptiles in private households over six years suggested a first-year rate of 75% [14] which highlights how these estimates can vary depending on source data.

Wildlife markets are one of the major acquisition channels for the modern exotic pet trade. They occur in several regions of the world where they take different forms. In North America and Europe, these “exotic pet expositions” typically involve indoor areas, and throughout the year the public may pay entry fees to gain access in many examples [15,16,17]. Proponents, organisers and sellers associated with wildlife markets have claimed that the animals kept and offered for sale at the events are not subject to stressful conditions [15]. Relatedly, proponents also claim that the temporary nature of the expositions (commonly one-day sales) means that the short-term housing and minimalistic provisions typically associated with these animals is acceptable [15].

Another major component of the modern exotic pet trade chain, at least in parts of the world where there is ready access to computers and the Internet, is a vast online culture of exotic pet videos and posts [18]. More than three billion people access, and are exposed to, content on social media every day (as of 2018, wearesocial.com), and the power of social media to influence public attitudes, consumer behaviour and lifestyle choices, including those relating to exotic pet ownership, is well recognised and references therein [19,20]. However, the posting of content involving exotic pets and the conditions in which they are being privately kept by enthusiasts and commercial breeders, also provides a growing unique opportunity to observe and assess the animal husbandry and potential animal welfare impacts of exotic pet ownership.

A poster child of the modern exotic pet trade, the Ball (or Royal) python (*Python regius*, family Pythonidae), a species distributed in western and central Africa, is the single most traded live animal legally exported from Africa under the Convention on International Trade in Endangered Species of Wild Fauna and Flora (CITES) [21]. The popularity of this species in the US and EU partly arises from its relatively docile nature, and the misconception that they require little specialised care [22]. Much of this international trade can be traced back to a number of registered reptile “farms” that are in operation across West Africa, most notably Benin, Togo and Ghana [13,23,24]. However, a significant proportion of the captive-bred reptile industry is based on the development of novel colour/pattern strains (also known as morphs) through artificial breeding selection [25].

International global regulations regarding the specific animal husbandry requirements for the private ownership and commercial captive breeding of reptiles, including the Ball python, are currently lacking. However, the welfare of reptiles is considered in some key pieces of national legislation, for example in England and Wales the Animal Welfare Act (2006) (under which all vertebrate species are covered). NGOs such as The Royal Society for the Prevention of Cruelty to Animals (RSPCA) also provide a number of key albeit non-binding recommendations. These include that Ball pythons should be provided with: (1) a vivarium that allows them to fully stretch out (i.e., with an enclosure at least as long as the snake’s total length, and a width and height being equal to at least a third of the snake’s length); (2) multiple shelters within their enclosures that provide them with the opportunity to hide; (3) a water bowl that is large enough to allow them to bathe fully submerged; and (4) appropriate substrate that allows them to maintain hygiene levels, and to express burrowing behaviour [26], and this guidance is broadly well supported in the objective scientific literature [27,28,29].

Our research focused on two components of current trade in Ball pythons as exotic pets described above: (1) exotic pet expositions; and (2) videos shared on social media platforms by Ball python breeders/sellers. Specifically, in this study we aimed to assess the housing conditions of Ball pythons at exotic pet expositions across Europe and North America, and those snakes being kept in “rack systems” as shared on the social media platform “YouTube”. We also aimed to quantify the number of vendors who provided husbandry information to potential Ball python consumers, both at exotic pet expositions and online via their public websites or social media platforms. Herein, it is hoped that our study will provide new insights into the animal welfare implications associated with the live trade in one of the most commonly traded reptile species involved in the global exotic pet industry, and that it can inform future management strategies and legislation.

## 2. Materials and Methods

### 2.1. Exotic Pet Expositions

We visited six representative expositions in North America (*n* = 3) and Europe (*n* = 3) using teams composed of between two and five researchers (Table 1) between 24.03.18 and 24.06.18. We assessed the housing conditions of Ball pythons therein using five environmental scoring criteria based on RSPCA minimum guidelines (Table 2). We assessed housing conditions of all Ball pythons on public display based on factors that did not require physical contact with the animals, and that could be quickly observed in a relatively short period of time (i.e., between 3–5 min). Prior to visits, we performed a series of inter-observer tests using a selection of images taken at expositions. The results of these reached a minimum of 90% agreement. If more than one snake were present in an enclosure, then we scored each snake separately. We recorded data on mobile phones and entered the data into an excel sheet on the same day (within hours after leaving the exotic pet exposition). In addition to scoring the housing conditions the following data was gathered:-Time of observation.-Vendor name (which was subsequently anonymous) for each stall selling Ball pythons (if advertised).-Total number of Ball pythons on display at each stall.-Number of snakes in each enclosure.-Environmental enrichment such as branches provided (Y/N) for each enclosure.-Whether the enclosure was transparent (Y/N).-Substrate type (if present) for each enclosure.-Ball python husbandry information provided by vendor via leaflets (Y/N).-Whether the Ball python was a “morph” (Y/N).

### 2.2. YouTube Videos

We accessed the social media platform YouTube on 12 July 2018 for videos that featured Ball pythons being kept privately by commercial breeders for onwards sale as exotic pets using the search terms “breeding Ball pythons”, “breeding royal pythons”, “Ball pythons for sale” and “royal pythons for sale” and created a video (URL) to facilitate future analysis. A single researcher systematically reviewed each of the videos on the list (*n* = 614) between 18 July 2018 and 30 September 2018. We excluded any duplicate videos (*n* = 137) that were obtained as a result of the different search terms used.

Our study focused on commercial breeders using rack systems (i.e., typically a multi-unitised system of plastic drawers in which individual tubs are inserted into the frame but do not close completely at the top; thus providing a degree of ventilation (see http://www.arscaging.com)). We excluded videos where Ball pythons were housed in a vivarium and/or other housing systems. We also excluded videos where one or more Ball pythons were not visible in a tray and thus a full housing assessment could not be made (e.g., if they were only observed being handled) (*n* = 364). In total, we excluded 501 videos from, and included 113 videos in the subsequent analysis.

Housing conditions for Ball pythons featured in videos were assessed using the same methodology and criteria used for snakes at the exotic pet expositions (Table 2). We assessed the housing conditions of each snake visible in a video, and we scored each snake separately if more than one snake was present. In addition to scoring the housing conditions, we also gathered data on:-The number of Ball pythons observed in each enclosure.-Whether the enclosure was provided with environmental enrichment (e.g., branches) (Y/N).-Whether or not the enclosure was transparent (Y/N).-Substrate type (if present) for each enclosure.-An estimate of the total number of enclosures observed in each video.-Whether or not the Ball python was a morph (Y/N).

To gain an insight into audience attitudes and information about the Ball python breeders/sellers, we recorded the name of the uploader (which was subsequently made anonymous), country of upload (where available), presenter gender, total number of comments (the first 10 in the list were categorised into positive, neutral and negative), number of “Thumbs up” and “down”, and finally whether morphs and/or genetic selection were verbally promoted in the video by the presenter (Y/N).

Finally, we selected the 10 most popular Ball python videos (according to the number of views). For each of the videos selected, we extracted the full text of all comments from the 10 “top comments”, as provided using the YouTube “sort by” tool and exported the text for each video to a text file for further analysis. Comment text was cleaned prior to analysis by removing symbols, numbers and transforming the text to lower case. Within the cleaned comment text, we identified the 10 most frequently appearing words as an indicator of comment content and a reflection of the sentiments of commenters. We visualised comment content using the “Jason Davies” package (www.jasondavies.com).

### 2.3. Educational Information: On-Line Review

Given its current position as the largest social media platform in the world, one researcher searched online between 9 February 2018 and 30 September 2018 for any commercial websites and social media pages provided via the social media platform “Facebook”. We specifically searched for online presence relating to vendors that were observed selling Ball pythons at one or more of the six exotic pet expositions visited in North America and Europe (Table 1) using commercial business names as search terms. When an online presence was observed, we assessed the presence/absence of any husbandry related information. We prioritised and only assessed husbandry information provided on commercial websites for any vendors who also had social media pages on Facebook (because we assumed that more detailed information would be provided on the former and would be where a potential consumer would expect to find such information).

We reviewed all text provided on vendor websites. For Facebook pages we reviewed all text provided on the “home” page, the “about” page, and all text in posts made during 2018. We translated text into English using Google Translate for information that was provided in other languages. In total we reviewed information provided by 57 different Ball python vendors on their websites (*n* = 26) and Facebook pages (*n* = 31). We found that 21 of the vendors observed at one or more of the six exotic pet expositions in North America and Europe did not have any observable online presence and therefore could not be reviewed. In particular we searched for any information relating to: (1) Ball python lifespan and size; (2) recommendations on feeding and water requirements; (3) conditions within the vivarium (including humidity, temperature and lighting); (4) spatial requirements; and (5) any recommendations on environmental enrichment and shelter.

### 2.4. Statistical Analysis

We carried out all statistical analysis using R statistical software version 3.4.1 (R Development Core Team 2017). We used Chi-square goodness of fit test to determine if the frequency of environmental score was distributed similarly for each environmental housing condition (mobility, shelter, water, substrate, hygiene). To test for an association between substrate and hygiene score a Chi-Square test association was used. To avoid expected values less than one, the two below adequate/acceptable scores were grouped (creating below and above adequate/acceptable levels). Where this analysis was not possible, a Fisher’s Exact test was applied. In order to analyse public perceptions of Ball python YouTube videos, a “Thumbs up” and “Thumbs down” score was created by dividing the number of “Thumbs up” and “Thumbs down” by the total number of views (to take into account some videos being available for longer periods of time, see [19]). A Generalized Linear Model (GLM) with a binomial distribution and then a quasi-binomial distribution was fitted to the data (due to over-dispersal) [30]. GLMs were applied separately to the “Thumbs up” score (number of “Thumbs up” given weighted by the number of views), “Thumbs down” score (number of “Thumbs down” given weighted by the number of views) and to the proportion of positive comments in the first 10 videos. Not all views generated a “Thumbs up”, “Thumbs down” or comment so each element was analysed separately. Residuals and validation plots were checked for overall model fit.

## 3. Results

### 3.1. Exotic Pet Expositions

In total we assessed the housing conditions of 4855 individual Ball pythons observed across six different exotic pet expositions in North America and Europe in 2018 (Deland, USA; *n* = 286); Doncaster, UK; *n* = 427; Houten, Netherlands; *n* = 523 & *n* = 1081; Madrid, Spain; *n* = 221; Tampa, USA; *n* = 1078; and Toronto, Canada; *n* = 1245). The mean number of snakes per enclosure was 1.03 snakes (min = 1, max = 4), and 37.36 per vendor (min = 1, max = 332). All specimens we observed received mobility, shelter and water scores below the minimum requirements set by the RSPCA (Score 1 and 2; Figure 1 and Figure 2) with all snakes being kept in enclosures smaller than the length of the snake, and with no shelter or water provisions. We found that substrate was either absent or inadequate for 76% of snakes observed (Figure 1 and Figure 2), with a significantly higher frequency of inadequate conditions than expected given an even distribution (χ^2^ = 2057.2, df = 2, *p* <0.001).

A significant negative association was found between hygiene score and substrate score (χ^2^ = 18.7, df = 1, *p* < 0.001), with a higher than expected number of enclosures observed as scoring well for both hygiene and substrate, as well as those scoring poorly for both hygiene and substrate. The lowest mean hygiene score was found when there was no substrate present, with the absence of substrate being the only type recorded with hygiene scores of 1 (i.e., unacceptable level of hygiene; detritus liberally present). We found a significant difference in the type of substrate provided (χ^2^ = 184.25, df = 12, *p* < 0.001), with wood shavings (35%) being the most common substrate type [followed by paper towel (25%), no substrate (11%) and reptile carpet (11%)]. Less than 0.1% (four Ball pythons) had access to any environmental enrichment and none of the vendors (observed selling Ball pythons) provided any information on Ball python husbandry. In contrast, we found that 99% of Ball pythons observed received a hygiene score that met the minimum RSPCA requirements (Score and 3; Figure 1 and Figure 2). At the exotic pet expositions where morph type was recorded, 94% (*n* = 2792 of 2984) Ball pythons observed were advertised as captive- bred morphs rather than wild-type individuals.

### 3.2. YouTube Videos

In total, we assessed the housing conditions of 787 Ball pythons (from a total of 1113 snakes present in the videos) featured in the 113 YouTube videos included in our study. The mean number of snakes per enclosures was 1.4 individuals (min = 1, max = 5), with the estimated number of trays per video ranging from < 20 to over 200. All of the snakes observed received mobility scores below the maximum standards (Score 1 and 2; Figure 3 and Figure 4), with 97% of snakes being kept in enclosures that were smaller than the total length of the snake. We also found that 96% of Ball pythons observed received scores relating to water and shelter access that was below the minimum requirements set by the RSPCA (Score 1 and 2; Figure 3 and Figure 4; shelter: χ^2^ = 1299.8, df = 2, *p* < 0.001; water: χ^2^ = 960.1, df = 2, *p* < 0.001). Of the Ball pythons we observed, less than 0.4% (three snakes) had access to any environmental enrichment.

In contrast, we found that a relatively higher proportion of the Ball pythons received scores relating to substrate (84%; *n* = 661) and hygiene (80%; *n* = 628) that met the minimum requirements set by the RSPCA (substrate: χ^2^ = 930.2, df = 2, *p* < 0.001; Hygiene: χ^2^ = 767.8, df = 2, *p* < 0.001). A significant association was found between hygiene score and substrate score (χ^2^ = 63.9, df =1, *p* < 0.001), with a higher than expected number of enclosures observed as scoring well for both hygiene and substrate, as well as those scoring poorly for both hygiene and substrate. We found a significant difference in the frequency of types of substrate used, with wood shavings (67%) again being the most common substrate type observed (χ^2^ = 791.2, df = 3, *p* < 0.001). Less than 1% (0.38%, *n* = 3) of snakes had access to environmental enrichment and 52% (*n* = 412) of snakes were in transparent enclosures. Of the Ball pythons featured in the videos, 98% (*n* = 772) were described as captive-bred morphs rather than wild type individuals.

We found that only six (5%) of the 113 videos assessed featured at least one negative comment within the first 10 comments posted (i.e., those which referenced concerns relating to observed low animal welfare scores). In contrast, we found that 98 (87%) of the 113 videos assessed featured at least one positive comment within the first 10 comments posted. The variables used to predict the number of “Thumbs Up” and “Thumbs down” (score weighted by the number of views) and the proportion of positive comments (within the first 10 comments) are shown in Table 3. We found that the number of “Thumbs up” increased significantly with increased level of hygiene and with the number of snakes in the enclosure, while the number of “Thumbs up” decreased significantly with increased substrate suitability (Table 3). The number of “Thumbs down” significantly decreased with the presence of transparent enclosures (Table 3). None of the predictors had a significant effect on the proportion of positive comments in the first 10 videos posted (Table 3).

With regards to word frequency for the 10 most watched Ball python videos, a total of 954 words were extracted and included in subsequent analysis. Comments were characterised by a number of “positive” terms, with the 10 most commonly used words being: “video” (*n* = 18), “like” (*n* = 17), “great” (*n* = 15), “love” (*n* = 12), “thanks”(*n* = 12), “see” (*n* = 10), “good” (*n* = 10), “videos” (*n* = 9), “would” (*n* = 9), and “get” (*n* = 8) (Figure 5).

### 3.3. Educational Information Online

In total we reviewed the husbandry information provided by 57 vendors on websites (*n* = 26) and Facebook pages (*n* = 31). We found that only eight (14%) of these vendors provided any information regarding Ball python husbandry (provided via seven websites and one Facebook page respectively). Three of these vendors were based in Canada (5%), two were based in the Netherlands (4%), two were based in the USA (4%), and one was based in the UK (2%). No significant association was found between the presence of online husbandry information and the region (North America vs. Europe) (Fishers Exact test, *p* > 0.05). We found that two vendors (4%) provided information regarding Ball python potential life span and size in captivity; three vendors (5%) provided information regarding how often a Ball python should be fed in captivity; seven vendors (12%) specifically stated that Ball pythons should be provided with a vessel of water for drinking and five vendors (9%) stated that the vessel should also be large enough to enable the snake to bathe.

We also found that four vendors (7%) provided information on vivarium humidity; none of the vendors provided any information on vivarium lighting; seven vendors (12%) provided information on vivarium temperature (all of which recommended providing a “hot” and a “cool” end); one vendor (2%) recommended providing environmental enrichment that would enable snakes to exhibit climbing behaviour; five vendors (9%) recommended providing shelters that were large enough for the snakes to hide under [three of which (5%) recommended providing a shelter at both the “cool” and “hot end”; and two of the vendors (4%) recommended that the snakes should be provided with a vivarium that allows the snake to stretch out to full length.

## 4. Discussion

### 4.1. Main Findings

Our study represents the largest and most in-depth review of the housing conditions provided by Ball python breeders and vendors carried out to date. It is clear that Ball pythons remain a commonly exploited species that is readily available for purchase predominantly in North America and Europe at exotic pet expositions and online via commercial websites and social media pages via Facebook. Despite this widespread availability, we found that the majority of entities involved in this commercial enterprise are not providing housing conditions that meet the minimum welfare recommendations provided by the RSPCA [26] and others, either in public (at exotic pet expositions) or privately (in rack systems prior to sale) for periods of time that could range from several days to years. Our study also highlights that overall, most entities are not providing detailed written guidance regarding husbandry to potential Ball python consumers, either at exotic pet expositions, or on their commercial website, or on any of their associated social media pages. Brief summaries of the potential impacts on Ball python welfare for each of the housing assessment criteria utilised in this study are provided below.

### 4.2. Mobility

We found that breeders and vendors typically utilised display and rack systems for Ball pythons that involved small, and typically highly restrictive enclosures, with dimensions that prevented occupants from adopting straight line body postures (i.e., extending their bodies to full and unrestricted natural length), or any full movement at all. Detailed studies focused on Ball python activity levels and home ranges in the wild are currently lacking. Compared with some snakes, Ball pythons are considered to be relatively sedentary in nature [31,32]. This has led some governmental guidance to advise that enclosures less than the total length of the snake are consistent with their welfare [33,34]. Other guidance suggests that certain “active” snakes require enclosures longer than their full body length, whereas more “sedentary” species do not [35,36].

However, numerous scientific and other reports emphasise that snakes, including more sedentary species such as the Ball python, require the ability to fully straighten their bodies to satisfy their need for behavioural normality, exercise, avoidance of stress and disease, alleviation of physical discomfort, and achievement of physical comfort [26,29,37,38,39] For example, Kreger and Mench [40] found that Ball pythons that were restrained in a container prior to handling demonstrated a significant rise in plasma corticosterone (CS) levels, potentially indicating an acute stress response. Given that the majority of the snakes observed during our study were not given the ability to extend their bodies to full and unrestricted natural length, it is arguable that the current husbandry practices could likely be having a negative impact on the welfare of the Ball pythons that are housed in this manner.

### 4.3. Shelter and Water

The Ball python is commonly referred to as a relatively “docile” species, due in part to its tendency to curl up tightly into a ball rather than to try and bite when handled [41]. Yet, such “head-hiding behaviour” i.e., the deliberate seclusion of head including under its own body, or objects and substrate is considered to be a response that can be related to fear, defense, and/or stress inducing experiences resulting from inappropriate captive environments (e.g., excess/rough handling and inadequate lighting [29,42]. It is generally advised that snakes, especially those that are considered reclusive in nature, should be provided with cover such as multiple hides that allow them to exhibit stress avoidance behaviour [26,41,43]. In light of these recommendations, our study suggests that the lack of adequate shelter provided by entities in display and rack systems (even if only temporarily, see below) could likely be having a negative impact on the welfare of the Ball pythons that are housed in this manner.

Similarly, we found that none of the Ball pythons at exotic pet expositions were provided with water containers, and that the majority of snakes observed in YouTube videos were not provided with water container sizes that met the guidance provided by the RSPCA [26] and others. Snake species differ in both drinking kinematics and water inflow patterns [44], and detailed studies for Ball pythons appear to be currently lacking in this regard. However, it is generally advised that Ball pythons should be provided with a vessel of clean fresh water for drinking that is replaced daily [26,41,43]. The provision of water vessels serves a dual purpose in that it also provides individuals with the opportunity to engage in bathing behaviour that can prevent negative welfare impacts (e.g., facilitation of normal skin sloughing and maintenance). Although there are some disagreements as to how important such behaviour is for captive Ball python welfare [43], arguably the potential negative impacts of current arbitrary water provision on Ball python welfare should not be ignored.

### 4.4. Hygiene and Substrate

In contrast to the other housing scoring criteria, we found that breeders and vendors typically utilised display and rack systems for Ball pythons that met the minimum hygiene standards as recommended by the RSPCA [26] and others (i.e., that they were observably “clean” with detritus being minimal or absent in the vast majority of cases). The fact that enclosures appear to have been regularly cleaned indicates some direct animal welfare benefits (e.g., by helping to mitigate the accumulation of particular potentially pathogenic wastes and microbes [29]. However, it is important to note that this cleanliness may have come at the expense of impacting negatively on other aspects of Ball python welfare. For example, vendors and breeders may have elected (intentionally or unintentionally) to provide relatively diminutive and barren enclosures in order to facilitate a regular cleaning regime.

With regards to substrate scores, in general, we found that the housing conditions observed online via YouTube videos largely met minimum RSPCA [26] recommended standards for this particular criterion. Yet, we found that this standard of care did not also extend to the Ball python housing conditions provided at exotic pet expositions, which largely did not meet minimum welfare criteria (e.g., Ball pythons had no substrates or coverage for <75% of their enclosure floor). As such, current substrate provision by vendors at exotic pet expositions could likely be having a negative impact on the welfare of the Ball pythons that are housed in this manner. In relation to substrate type, the fact that dry wood shavings were most commonly observed at both expositions and in YouTube videos is perhaps favorable. Wood shavings are generally considered to be a convenient artificial substrate consistent with good Ball python welfare in captivity [26] due to its ability to absorb moisture (that can help to prevent negative animal welfare impacts such as pneumonia [43]. However, it is important to note that in both cases other substrate types that are generally considered as poor substitute material (such as paper towel) were frequently observed.

### 4.5. Duration and Purpose of Captivity

Our study demonstrates that Ball pythons are being provided with unsuitable housing conditions for relatively long periods of time. The duration of time that Ball pythons were housed in the conditions observed in our study will likely have varied depending on whether they were being publicly displayed at an exotic pet exposition or being privately kept in a rack system elsewhere. However, the former can still constitute several days (taking into account both transport and exposition duration), and the latter can potentially involve far longer periods of time (e.g., months or years, if not the entirety of the snakes’ lifetime). It is reasonable to assume that the environmental conditions provided by breeders (as shared via social media) reflect how some vendors keep snakes between expositions. Whilst it is generally accepted that wildlife can be held under temporary conditions that would not be acceptable for longer-term accommodation (e.g., clinical and quarantine situations) [29], it is dubious as to whether commercial captive breeding and display for subsequent sale falls under these criteria.

### 4.6. Selectively Bred Morphs

We found that selectively bred Ball python morphs are predominantly being sold via vendors and breeders at exotic pet expositions in North America and Europe and advertised in online videos via YouTube (rather than “wild” snakes sourced directly from Africa). Theoretically, commercial captive breeding can help to protect wild populations in some situations by reducing unsustainable wild captures. However, concern and evidence exist regarding fraudulent claims where wild-caught animals are sold as captive-bred [45]; and Ball pythons are no exception [46]. Also, it has been argued that captive breeding generates various propagation-specific welfare problems and is not a harmless alternative to wild capture [12,47].

There is a growing awareness and concern regarding several genetic disorders associated with artificial selection for colour and pattern morphs, and the negative impacts that this may be having on the health and welfare of Ball pythons [25]. For example, “wobble head” syndrome is a central nervous system disorder that occurs primarily in “spider” morph ball pythons [48]. Reported clinical signs include side-to-side head tremors, incoordination, erratic corkscrewing of the head and neck, inhibited righting reflex, torticollis, poor muscle tone, and loose grip with the tail [25].

Some breeders perceive affected animals as having a reasonable quality of life because they will continue to feed and breed in captivity. However, feeding and breeding activities have been concluded to be unreliable indicators of welfare state in animals generally [49] including both reptiles [50,51] and snakes specifically [50,51]. Further, it is reported that individuals are euthanised in severe cases as their feeding response and strike accuracy is very poor due to incoordination [25]. This syndrome is directly related to selective breeding choices, because there is a well-characterised relationship between the spider morph and “wobblers” [25]. Other common conditions include deformities of the spine, e.g., “kinks” [52], skull, e.g., “duckbills” and eyes, e.g., “bug eyes” [25,53]. Although detailed studies focused on the welfare impacts of these traits are currently lacking, there are potential implications for Ball python welfare (e.g., feeding and respiratory problems) that need to be explored.

### 4.7. Education and Consumer Awareness

Our study found that the vast majority of vendors did not provide any written Ball python husbandry information (consistent or inconsistent with RSPCA recommended guidance [26]) to potential consumers either at exotic pet expositions, or on their commercial websites, or on their social media pages via Facebook. Furthermore, we found that only six (5%) of the 113 YouTube videos assessed featured any negative comments within the first 10 comments posted, and that the most frequently used terms appeared to be positive in nature, despite receiving poor scores for mobility and shelter criteria. As such, our study also indicates that at least some potential consumers of Ball pythons as exotic pets may themselves be unable to recognise unsuitable housing conditions that do not meet minimum animal welfare standards. If potential Ball python consumers are purchasing Ball pythons without full consideration of the responsibilities associated with ownership, then this could be having long-term negative animal welfare impacts (e.g., via inappropriate care and abandonment at rescue centres).

## 5. Limitations

We recognise that wild animal husbandry best practice constitutes an ever-evolving field of research and that guidance on how best to maintain animal welfare standards can differ greatly between sources and between countries in some cases dependent on an entities role in the exotic pet industry. In particular non-scientific “folklore” guidance has been noted as a major source of utilised information that can have unintentional negative impacts on the welfare of captive animals including exotic pets [29,54]. It is important to note that it was beyond the scope of our study to compare how guidance on Ball python husbandry has differed over time or between sources. However, in the absence of independent internationally recognised standards, we chose the RSPCA UK guidelines as the basis of our housing assessment criteria because they were developed and recently updated by a long-standing NGO whose stated public aim is “ensuring that every pet is cared for properly and has a good home”, and because, as stated previously, this guidance is broadly well supported in the objective scientific literature.

Our study was necessarily limited (by factors such as time, resources and access) and could not be exhaustive. Specifically, we restricted our welfare assessments to six exotic pet expositions located on two continents and to 113 videos identified via search terms in one language that were posted on one social media platform. Similarly, we restricted our welfare assessment to a particular set of housing related criteria that could be carried out quickly and that did not involve direct handling, physical examinations or behavioural observations. Equally, we recognise that information on Ball python husbandry could be provided via other sources not examined in this study, e.g., verbally or in hobbyist magazines and books. However, in lieu of detailed readily accessible public information, we believe that this study presents preliminary data for over 4800 Ball pythons and represents one of the most comprehensive reviews focused on the animal welfare impacts of current practices in the exotic pet industry that has been carried out to date.

## 6. Recommendations

Given the current prevalence of reptiles at exotic pet expositions and in private ownership as exotic pets, more welfare-related research is required to investigate the impacts of this large-scale commercial practice. Our study provides a useful insight into the impacts of this industry on one of the most commonly utilised species (the Ball python) that can help to guide such future research effort. In particular we recommend that future initiatives look to expand on the welfare criteria utilised in this study to also incorporate other housing criteria (e.g., light, temperature, humidity and ventilation), physical examinations and behavioural criteria. Research to further compare welfare conditions at a broader range of expositions and to assess the welfare impacts of non-commercial private owners should be considered. Research focused on the welfare impacts of the artificial selective breeding of morphs is also recommended. Such information could help inform a range of different operational initiatives aimed at reducing negative animal welfare impacts, including improved husbandry and policy change. Arguably this data should be based upon increased understanding of the natural history of this species including its behaviour and biology in the wild.

Research (gathered through interviews and focus group discussions) could aim to understand Ball python breeders/sellers and keepers motivations for acquiring this species and also their perceptions and beliefs around reptile sentience and welfare, two issues of increasing relevance [55]. It would also be valuable to explore the common health problems experienced by Ball pythons when brought to veterinarians by their owners. This information could be gathered by surveying veterinary clinics. Such information could aid consumer behaviour change programmes, which aim to reduce the demand for exotic pets [56] and also aid the development of educational materials. In this study we found that 100% of the YouTube sellers/breeders were male, further demographic information could be gathered that could further inform tailored interventions. We suggest that more consistent guidance, adherence to agreed husbandry principles, and more potent operating models that are formally incorporated into relevant legislation would greatly aid existing and future efforts to safeguard animal welfare in this regard.

## 7. Conclusions

Many elements of commercial trade in reptiles as exotic pets are known to compromise both their physical and psychological welfare including transportation, handling, storage, intensive captive breeding, subsequent captivity stress, injury and disease [12,57]. Despite the large numbers being kept in captivity, and the demand for novel morphs, information about the health and welfare of Ball pythons (and indeed other reptiles) in this context have received little attention within the published literature [25,58]. It has also been suggested that exotic pet expositions set a poor example of animal husbandry that may be adopted by purchasers [15]. Our study represents the largest and most in-depth review of the housing conditions provided by Ball python breeders and vendors carried out to date. It is hoped that this research serves to stimulate new research focused on Ball pythons, both in the wild and in captivity, which can be used to inform husbandry guidelines, policy improvements and consumer awareness initiatives.

## Figures and Tables

**Figure 1 animals-10-00413-f001:**
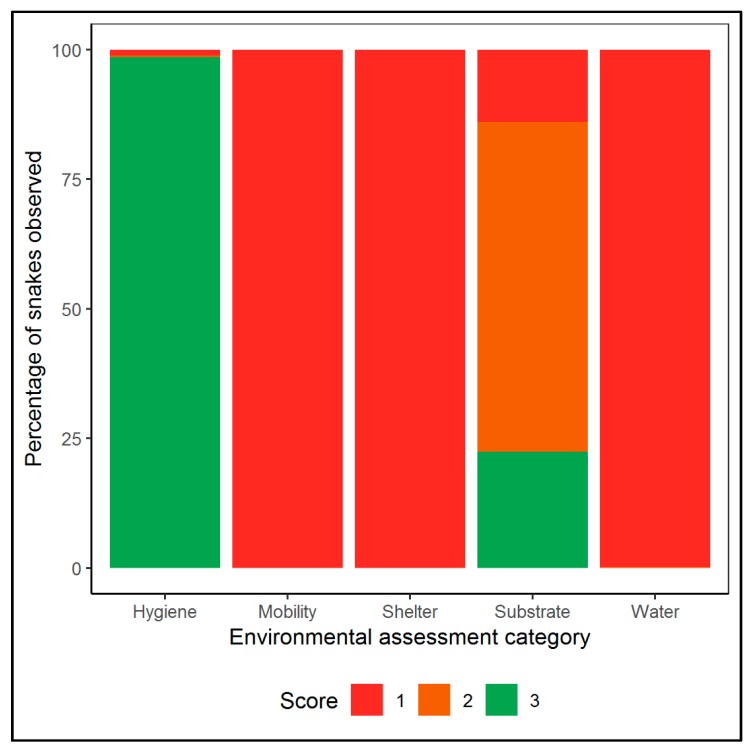
Housing assessment for the 4855 Ball pythons observed at exotic pet expositions in North America and Europe during the study. Criteria score (see Table 2) is based on the minimum requirements recommended by the RSPCA and others (1–2 = below minimum requirements; 3 = adequate i.e., minimum requirements met).

**Figure 2 animals-10-00413-f002:**
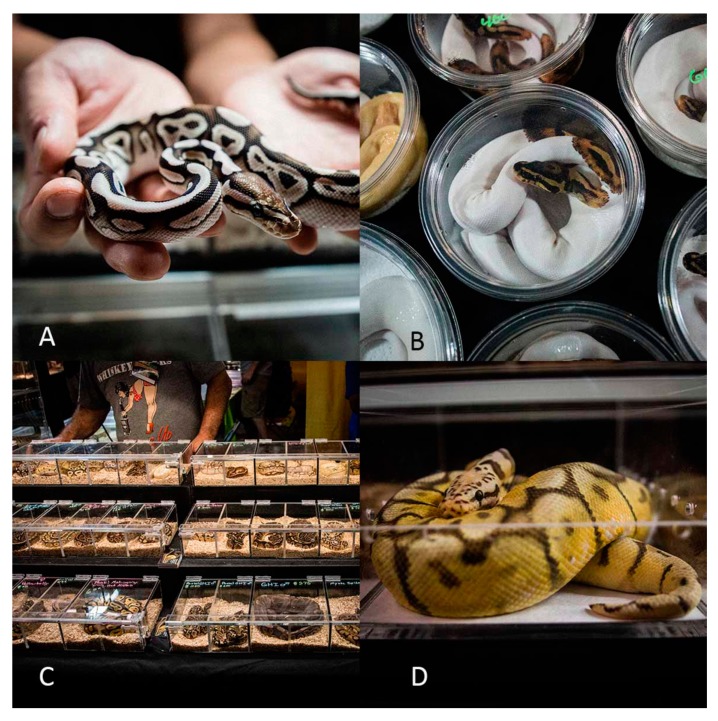
Images taken from an exotic pet exposition in the USA that reflect housing conditions commonly provided. (**A**): A Ball python morph being handled by a vendor; (**B**): A Ball python morph offered for sale in a small plastic container; (**C**): A number of Ball python morphs offered for sale in a plastic display rack; (**D**): A Ball python morph offered for sale in a small plastic container. Images (**A**–**D**) © Neil D’Cruze.

**Figure 3 animals-10-00413-f003:**
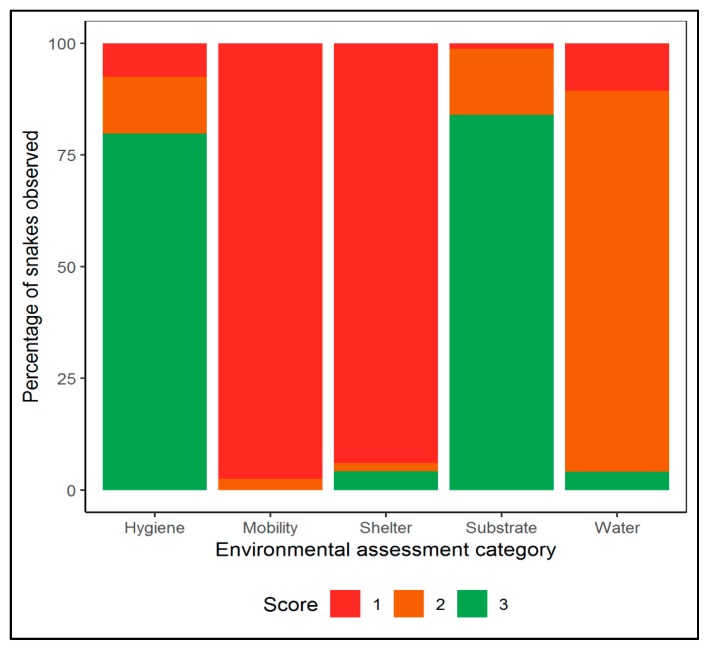
Environmental assessment for the 787 Ball pythons observed online in 113 YouTube videos. Criteria score (see Table 2) is based on the minimum requirements recommended by the RSPCA and others (1–2 = below minimum requirements; 3 = adequate i.e., minimum requirements met).

**Figure 4 animals-10-00413-f004:**
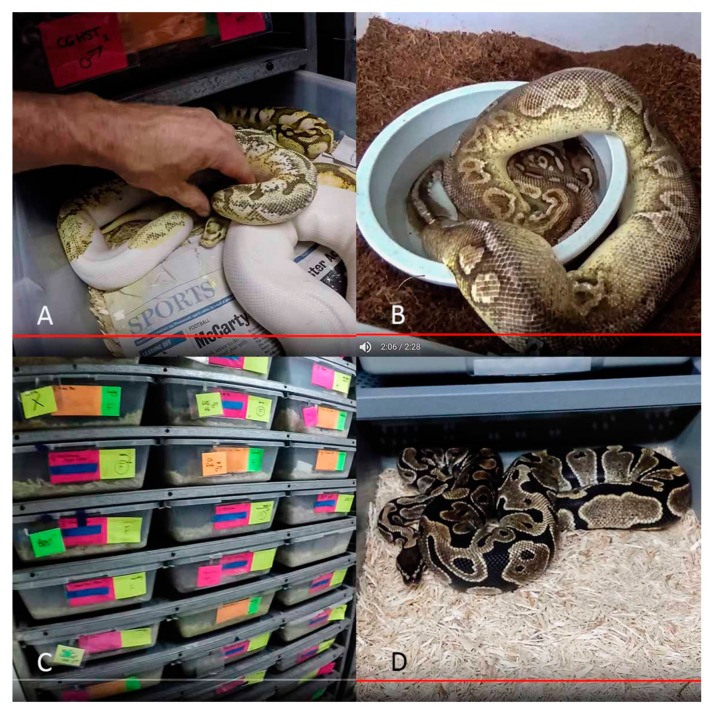
Images taken from YouTube videos assessed during this study that reflects housing conditions commonly provided. (**A**): A Ball python morph being handled by a breeder; (**B**): A Ball python morph intended for sale housed in a plastic container; (**C**): A number of Ball python morphs offered for sale in a plastic display rack system; (**D**): A Ball python morph offered for sale in a plastic container. Images (**A**–**D**) © YouTube.

**Figure 5 animals-10-00413-f005:**
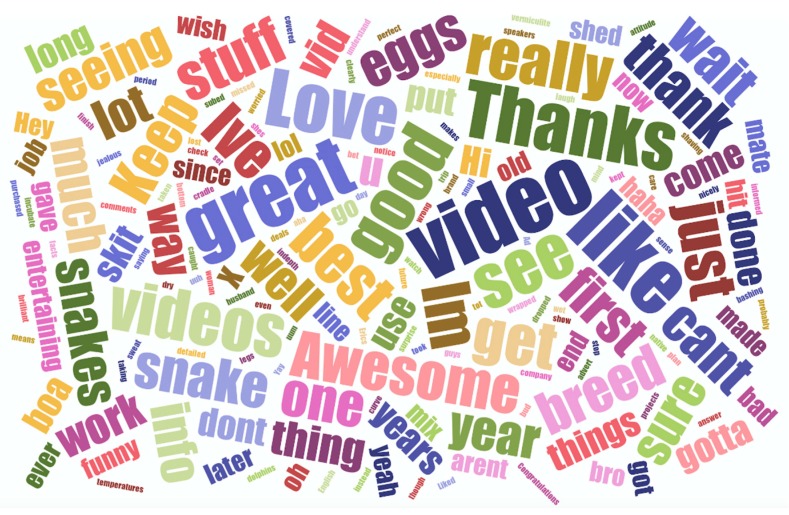
Wordcloud depicting the nature of comments for the 10 “top comments” from the 10 most watched Ball python videos. Font size is relative to the frequency at which each of the words appears in the comments.

**Table 1 animals-10-00413-t001:** Organiser, location, date of visit, and number of researchers who attended each pet exposition. Total number of snakes = 4861. Please note that six enclosures had two snakes where only one snake was assessed and included in our environmental scoring criteria.

Exposition Organiser	Location of Exposition	Date of Visit	No. Vendors	No. Snakes	No. Researchers
Madrid Expo Terraria	Madrid, Spain	24.03.18	9	221	5
Canadian Reptile Breeders’ Expo	Toronto, ON, Canada	06.05.18	28	1245	2
Terraria Houten	Houten, The Netherlands	03.06.18	15	523	3
Repticon Tampa	Tampa, FL, USA	09 & 10.06.18	24	1078	4
Repticon Deland	Deland, FL, USA	16.06.18	7	286	3
Expo Terra Doncaster	Doncaster, UK	24.06.18	28	427	2
Terraria Houten	Houten, The Netherlands	23.09.18	31	1081	1

**Table 2 animals-10-00413-t002:** Ball python environmental assessment criteria, this was developed based on the minimum requirements recommended by the RSPCA [26,27,28,29].

No.	Category/Score	1	2	3
1	Mobility/Space	Length of enclosure = less than length of snake.	Length of enclosure = length of snake.	Length of enclosure = approx. 1.5 X length of snake.
2	Shelter	No shelters present.	Shelter present, which does not cover 100% of snake when coiled up.	Shelter present, which covers 100% of snake, when coiled up.
3	Water	No water present.	Clean water available. Water bowl is too small to allow the snake to soak its entire body.	Clean water available. Water bowl is large enough to allow the snake to soak its entire body.
4	Substrate	No substrate present (covers 0% of enclosure floor).	Inadequate substrate present (covers < 75% of enclosure floor).	Adequate substrate present (covers > 75% of enclosure floor)
5	Hygiene	Unacceptable level of hygiene. Detritus liberally present.	Intermediate between 1 and 3.	Acceptable level of hygiene. Detritus minimal present/absent.

**Table 3 animals-10-00413-t003:** Model parameters for quasi-binomal generalised linear models (QBGLM) on the characteristics of Ball python YouTube videos. Models used to assess audience responses to housing conditions. Analysis of negative comments using QBGLM was not carried out due to the small number of videos with negative comments (5%—six of 113 videos).

Response Variable	Fixed Factor	Estimate	SE	T–Value	*p*–Value
Thumbs Up Score	Space	−1.207	1.056	−1.143	0.256
	Shelter	0.204	0.311	0.654	0.514
	Water	0.159	0.171	0.925	0.357
	Substrate	−0.687	0.118	−5.812	<0.001
	Hygiene	0.502	0.132	3.802	<0.001
	Number of Snakes	1.192	0.150	7.924	<0.001
	Transparency	0.093	0.160	0.579	0.564
Thumbs Down Score	Space	0.825	0.720	1.145	0.255
	Shelter	−0.994	1.205	−0.825	0.411
	Water	0.068	0.328	0.208	0.835
	Substrate	0.025	0.224	0.112	0.911
	Hygiene	−0.182	0.208	−0.875	0.383
	Number of Snakes	−0.168	0.294	−0.572	0.568
	Transparency	−0.658	0.311	−2.117	0.037
Proportion of Positive Comments Within First 10 Videos	Space	−1.230	1.784	−0.690	0.492
	Shelter	0.020	0.329	0.062	0.951
	Water	−0.099	0.416	−0.237	0.813
	Substrate	0.119	0.339	0.350	0.720
	Hygiene	0.040	0.241	0.166	0.869
	Number of Snakes	−0.306	0.282	−1.088	0.279
	Transparency	−0.363	0.289	−1.255	0.212

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
