# Peer review of "Dropping the Ball? The Welfare of Ball Pythons Traded in the EU and North America"

_animals, 2020, doi:10.3390/ani10030413_

Round 1

Reviewer 1 Report

The study uses a comprehensive data set gathered by direct observation of snake vendors and a novel online set derived from YouTube and associated websites and social media. One can argue that vendors at pet shows contain and display the ‘product’ for the convenience of the seller (space limited) and the buyer (transport limited) and so conditions are rightly identified as welfare-poor and the issue is what time period is acceptable. Even so the authors rightly criticise vendors for not providing adequate information on the ideal conditions that the buyer should achieve for their ‘pet’. The dataset is reasonably analysed with some statistical rigour and comprehensively discussed. The latter might seem excessive but provides a wealth of information of specific and general interest.
The manuscript is well-written but a number of minor errors are identified in the following.
Line 39, 459: RSCPA to RSPCA
Line 87: Python regius in italics
Line 105: have an appropriate
Line 123, 130, 132, 235,262, 343, 379, 418, 489, 491: Inconsistent capitalisation of Ball in Ball python. These are some instances of ‘ball’ rather than ‘Ball’.
Line 247: XX snakes?
Line 282: for the ten most watched
Line 312, 318: Cannot find reference to these figures in text?
Line 454: entity’s
Line 480: repeat as a source

Author Response

Dear Reviewer 1,

We would like to thank you for your careful consideration of our manuscript, and your insightful feedback.

In particular we are pleased that you consider our manuscript to be based on a comprehensive data set that has been analysed with statistical rigour along with a comprehensive discussion of how our findings relate to the use of Ball pythons as exotic pets.

Please find an overview of the specific amendments that we made to the manuscript in response to your feedback. We have also reviewed the paper and made other small grammatical amendments that are visible via the tracked changes function. 

Should any thing further be required we will of course be happy to respond as required.

Best Regards,

Neil

Reviewer 2 Report

23-24     Include some taxonomic information (also 36-37)

25           Define “wild animals”

30           Explain the RSPCA to international readership

66-69     This is a wide variation – what is the significance of these different findings?

70           Poor English

87           Italics needed, and taxonomic information would be beneficial

88           What is CITES? What Appendix is the ball python listed on?

93           Captive-bred

98           The Animal Welfare Act 2006 only applies to England and Wales

99-107   Why were the RSPCA guidelines selected? The authors admit that there is little research on ball pythons in the wild (e.g. 342-343), so how can the guidelines be supported “in the objective scientific literature”?

103         Typo

105         Poor English

116         This wording implies that the species is common; surely not what is being suggested? It is common IN TRADE

120-125                How did you ensure observer / assessor consistency?

139         Table 2 – how did the observers define adequacy /acceptability? More information is needed on this point

146-147                Why these search terms and not others? E.g “pet ball python”?

144-156                These numbers are either unclear, or do not add up - ?

158-159                This risks introducing double-counting in the data, surely?

162         Define enrichment

183         Why just Facebook? Other platforms are commonly in use

211         State GLM in full

226-228                Not surprising, given the high scoring

247         “XX”? Is there a figure missing here?

282         What does this tell us?

442-443                Define negative – and why is this finding significant?

Author Response

Dear Reviewer 2,

We would like to thank you for your careful consideration of our manuscript, and your insightful feedback.

Please find an overview of the specific amendments that we made to the manuscript in response to your feedback. In particular you will note that we have addressed your queries regarding how we defined our substrate score, and also how we ensured assessor consistency. We have also reviewed the paper and made other small grammatical amendments that are visible via the tracked changes function. 

Should any thing further be required we will of course be happy to respond as required.

Best Regards,

Neil
